# Use of Cardio-Pulmonary Ultrasound in the Neonatal Intensive Care Unit

**DOI:** 10.3390/children10030462

**Published:** 2023-02-26

**Authors:** Elena Ciarmoli, Enrico Storti, Jessica Cangemi, Arianna Leone, Maria Pierro

**Affiliations:** 1Neonatal and Pediatric Unit, ASST Brianza, 20871 Vimercate, Italy; 2Department of Critical Care, Maggiore Hospital, 26100 Cremona, Italy; 3Neonatal and Pediatric Intensive Care Unit, Bufalini Hospital, AUSL Romagna, 47521 Cesena, Italy

**Keywords:** lung ultrasound, targeted neonatal echocardiography, cardiopulmonary ultrasound, neonates

## Abstract

Cardiopulmonary ultrasound (CPUS), the combination of lung ultrasound (LUS) and targeted neonatal echocardiography (TnECHO)AA, may offer a more appropriate approach to the challenging neonatal cardiovascular and respiratory disorders. This paper reviews the possible use of CPUS in the neonatal intensive care unit (NICU).

## 1. Introduction

Cardiorespiratory disorders are the major cause of death in the neonatal intensive care units (NICUs) [1]. Diagnostic and monitoring tools are paramount in order to improve the outcomes in sick neonates suffering from cardiorespiratory disorders. Unfortunately, most of the options available in the adult and pediatric intensive care units to guide mechanical ventilation and support unstable hemodynamics are neither available nor exploitable in the NICU. Lung ultrasound (LUS) and targeted neonatal echocardiography (TnECHO) are safe, repeatable, non-invasive tools that are being increasingly used at the neonatal bedside. Cardiopulmonary ultrasound (CPUS), the combination of LUS and TnECHO, may allow for a better understanding of the cardiorespiratory challenges that the neonatologist has to face on a daily basis, tailoring the diagnostic approach and targeting treatment. This paper aims to review the current knowledge and potential use of CPUS in neonates.

## 2. Targeted Neonatal Echocardiography

TnECHO, performed by the neonatologist, allows for a real-time assessment of the cardiovascular state, supporting clinical decision-making [2]. Provided that a complete structural heart assessment to rule out the presence of congenital heart defects (CHDs) remains a prerogative of the pediatric cardiologist, TnECHO has been validated for several point-of-care purposes, including: (i) the evaluation of cardiac function during cardiorespiratory instability, (ii) the presence and the hemodynamics effects of patent ductus arteriosus (PDA), (iii) the diagnosis, treatment and follow-up of neonatal acute and chronic pulmonary hypertension (PH), (iv) the detection and treatment of pericardial effusion, (v) the guidance of central line placement.

### 2.1. Technique and Training 

Over the past few years, significant effort has been put into the implementation of standard operating procedures to appropriately train clinicians in performing TnECHO [3,4,5,6]. According to the European guidelines [4], a basic training should cover a period of at least 6 months in a NICU and/or pediatric cardiology unit, performing at least 100 scans, of which 70% should be normal. The training for advanced evaluation of normal heart anatomy, PDA and its characteristics, PH with its hemodynamic implications is supposed to cover a 6–12 month period in a NICU and/or pediatric cardiology unit [4]. These training guidelines may improve inter-observer reproducibility, diagnostic accuracy and reliability, potentially ameliorating the overall patient outcome.

### 2.2. Patent Ductus Arteriosus Evaluation

Patent ductus arteriosus (PDA) is a frequent finding in the neonatal population. The impact of PDA on the premature circulation depends on the amount of blood passing through the duct itself, which is determined by the pressure gradient between its two extremities (aortic and pulmonary ends). In the case of a large shunt from the left to the right side of the heart, signs and symptoms related to pulmonary overcirculation and systemic hypoperfusion may develop. Pulmonary overflow is due to increased pulmonary venous return which in turn causes left atrial volume overload and subsequent left ventricular dysfunction [7]. Systemic hypoperfusion is caused by post-ductal stealing phenomena.

TnECHO enables the neonatologist to assess the hemodynamic significance of the PDA with a multiparametric approach [5,6,8,9]. In order to standardize the PDA evaluation, the echocardiographic examination should provide the following information [9,10]: (1) ductal dimension and characteristics (transductal patency and diameter, direction and velocity pattern of the shunt); (2) signs of pulmonary overflow (3) signs of systemic hypoperfusion (Appendix A, Appendix A) [10,11].

However, despite the great work and effort undertaken in the past decades to better define hemodynamic ductal significance, there is still no evidence that PDA treatment may improve the overall outcome [12,13]. The recent BeNeDuctus trial showed that the expectant management for PDA in extremely premature infants is non-inferior to ibuprofen therapy administered according to pre-specified echocardiographic criteria of hemodynamic significance [13].

### 2.3. Early and Late Pulmonary Hypertension

Pulmonary hypertension of the newborn is a condition characterized by increased pulmonary vascular resistances [14] that could affect full term newborns and premature infants at different stages of their life. Early and acute forms of PH recognize a pre-capillary source related to the following mechanisms: (1) “maladaptation” to extrauterine life with reactive pulmonary vasocostriction; (2) “maldevelopment” involving remodeling of pulmonary vasculature and (3) “underdevelopment” with variable degree of hypoplasia of pulmonary vessels. Late PH develop often in the context of the chronic lung disease of prematurity, known as bronchopulmonary dysplasia (BPD). BPD is associated with changes in the vascular and pulmonary lung compartments, predisposing to the development of high venous post-capillary pressure and subsequent PH [14].

TnECHO not only reliably documents the presence of early and late PH but also provides information on (1) PH severity, (2) burden on right ventricular performance and pulmonary hemodynamics, (3) consequences on left ventricular performance and systemic blood flow. Therefore, TnECHO allows for targeted treatment choices and monitoring of the therapeutical effects. 

Appendix A and Appendix A summarize the most important TnECHO parameters used in the evaluation of PH in the neonatal period.

### 2.4. Targeted Choice of Volume Replacement, Inotropes and Vasopressors during Neonatal Shock

Shock is a pathologic condition causing an imbalance between tissue oxygen delivery and demand. In the initial phase of the shock, the increased oxygen demand is compensated by a maximization in oxygen extraction through neuroendocrine mechanisms, that increase the heart rate and cause vasoconstriction in the non-vital organs. Therefore, blood flow and oxygen delivery to the vital organs are maintained at the expense of the non-vital organs. The compensated phase of the shock is characterized by subtle clinical signs, such as mild tachycardia, prolonged capillary refill time and decreased urine output, with normal blood pressure. If shock progresses, blood pressure drops and the compensatory mechanisms become inadequate to maintain perfusion to the vital organs. Following this phase, the shock becomes irreversible [15]. Hypovolemia, myocardial dysfunction and abnormal peripheral vasoregulation are the major determinants of neonatal shock. Hypovolemia is detected at TnECHO, combining multiple aspects, such as reduced left ventricular end-diastolic diameter, evidence of “kissing walls” (collapse of the left ventricular walls at end-systole to the intra-ventricular septum), reduced hepatic venous flow and collapsed vena cava (Appendix A). In ventilated infants, a distended inferior vena cava is not a reliable marker of volume status. High intrathoracic pressure (due to positive pressure ventilation) can hinder the venous return at the level of the inferior vena cava, showing a falsely well-filled vessel, when the cardiac chambers themselves are under-filled [16]. Reduced volume status needs to be corrected before attempting other treatments of neonatal shock. Right and left myocardial dysfunction is easily recognized at TnECHO, either with standardized measurements (Appendix A) or eyeballing (Appendix A). Detection of myocardial dysfunction at TnECHO in a patient with shock and normal volume status indicates the need for inotropic agents, such as dobutamine, epinephrine, milrinone. The choice of the specific inotropic agent depends on the underlying clinical condition and blood pressure (Figure 1). Abnormal vasoregulation in neonates can only be supposed based on clinical presentation and/or persistence of shock despite normal volume status and normal ventricular function. When considering abnormal vasoregulation, the treatment choice falls on vasopressors, such as dopamine, noradrenaline, vasopressin, terlipressin. Again, the choice of the specific inotropic agent depends on the underlying clinical condition (Figure 1).

## 3. Lung Ultrasound

### 3.1. Lung Ultrasound Technique and Training

As opposed to TnECHO, the learning curve of LUS is relatively shallow and the reproducibility is very high [17]. Guidelines for neonatal LUS standardization have been recently released [18,19]. LUS is performed using different probes, micro-linear, “hockey stick”, phased-array sectorial (6–12 MHz), micro-convex (8 MHz) and high frequency linear probes (6–12 MHz) with good inter-rater agreement and interpretation reliability, although non-linear probes show lower agreement in less experienced hands [20]. When performing LUS, each lung is divided into three zones (anterior, lateral and posterior), using the axillary lines as boundaries. Each zone is then distinguished into an upper and a lower part. For each region, longitudinal and transverse scans are performed by keeping the probe longitudinally or perpendicular to the ribs. Anterior and lateral fields are examined in the supine position, posterior fields in prone or side position [19]. Although a recent study, comparing prone versus supine position during LUS in neonates, demonstrated comparable results one hour after postural change [21], the operator should consider the posture kept by the newborn before the examination, as the dependent region may show signs of derecruitment and/or fluid buildup in the interstitial space.

### 3.2. Lung Artifacts

As opposed to most sonographic examinations, LUS is predominantly based on the analysis of ultrasound artifacts. In this paper we will refer to the usual lung ultrasound lexicon, that define the two major artifacts on LUS as A and B lines, although recent observations addressing the origin of the ultrasound lung artifacts [22,23] are changing the terminology according to the principles that govern the ultrasound propagation through the lung [24,25]. 

#### 3.2.1. A-Lines

The air-filled lung parenchyma cannot be directly visualized at LUS. In the normally aerated lung, when the ultrasound beams travel through the chest wall and hit the pleura, up to 99% of the beams are reflected back. [26]. The ultrasound waves keep bouncing back and forth, while their energy attenuates at each bounce, creating the horizontal equidistant artifacts that are called A-lines (Figure 2A).

#### 3.2.2. B-Lines

B-lines are vertical reverberation artifacts that arise from the pleural line. B-lines are directed below to the deepest part of the screen and move synchronously with the lung sliding [25] (Figure 2B). B-lines are caused by pleural acoustic traps. An acoustic trap is characterized by a transonic access to the sub-pleural enlarged interstitial tissue, with a variable shape and content (water, cells, and connective tissue), and surrounded by residual air [25,27,28,29].

### 3.3. Lung Ultrasound Patterns

Lung US patterns are based on the presence of specific lung artifacts and signs, forming a consistent and characteristic layout (Figure 2) [30,31,32]. LUS patterns can be classified in different ways. A useful classification of LUS patterns, with the corresponding lung aeration score, is the following: A-Pattern (Figure 2A), normally aerated lung (aeration score = 0): A-lines, bilateral lung sliding (Appendix A). The A-pattern in the absence of lung sliding suggests the presence of pneumothorax. In that case the lung point needs to be searched for. The lung point is point where the pneumothorax ends and the normal contact between the parietal and visceral pleura is restored. The corresponding ultrasound image is a scan where the a normal sliding is detected right aside a static pleura, representing the loss of contact between the parietal and visceral pleura. Therefore after the lung point, the pleural space is filled with air and the pleural sliding is absent (Appendix A);B1-pattern (Figure 2B), moderate loss of lung aeration (aeration score = 1): three or more B-lines occupying less than 50% of the scanned area;B2-pattern (Figure 2C), severe loss of aeration (aeration score = 2): coalescent B-lines, occupying more than 50% of the scanned area;White lung (Figure 2D): compact B-lines that cause the acoustic shadow of the ribs to disappear within the entire scanning zone, anteriorly and posteriorly without spared areas;C-Pattern (Figure 2E): complete loss of aeration (aeration score = 3) leading to lung consolidation. The C-pattern is the only LUS pattern based on the anatomical visualization of the lung. Lung collapse or consolidation remove the A-line artifacts and allow for direct visualization of the lung parenchyma. Lung consolidations (that can highly vary in size and number), generally appear as hypoechoic or “liver-like texture” images, with irregular margins, containing air bronchogram and a vascular pattern enhancing an intraparenchymal pulmonary shunt. Lung consolidation may be due to lung atelectasis, pneumonia or MAS (Appendix A).

### 3.4. Pleural Line and Distribution of the B-Lines

In adult patients, the presence of B-lines has a great sensitivity in picking up pathological processes involving the interstitium, although the specificity in distinguishing the possible underlying process (i.e., pulmonary edema, pneumonia, interstitial lung disease due to fibrosis) is still debated [33]. The evaluation of the B-line’s distribution and the pleural line characteristics may increase LUS efficiency in providing differential diagnoses for interstitial diseases. Studies enrolling adult patients proved that B-lines due to pulmonary fibrosis/inflammation generally start at the posterior lung basis and are often associated with irregularities of the pleural line and subpleural small consolidations [25]. In this pattern of lung injury, the distribution of artifacts is inhomogeneous: spared areas of normal sonographic lung appearance are usually surrounded by areas containing multiple B-lines and pleural line irregularities. In contrast, B-lines are usually detected in a more homogenous, gravity-related distribution, in cardiogenic pulmonary edema [34,35,36,37].

### 3.5. Lung Ultrasound Predictive Scores

Predictive LUS scores can be performed with different methods, that have been recently revised elsewhere [30]. Studies suggest that LUS scores are predictive of disease severity, and needed for surfactant treatment and nasal continuous positive airway pressure (NCPAP) failure in preterm infants suffering from respiratory distress syndrome (RDS) [31]. LUS scores can foresee readiness to wean off NCPAP in premature infants with evolving BPD [38].

A multi-centric study involving 240 preterm neonates showed that a cut-off of 9 on LUS score was predictive of the first surfactant administration [39]. De Martino and colleagues, found a good sensitivity and specificity of the LUS score to predict the need for surfactant therapy during NCPAP in extremely preterm neonates with RDS, with a cut off value between 6 and 8 [40].

Serial monitoring of lung aeration with LUS has demonstrated a good predictive value in stratifying preterm infants at low and high-risk of developing BPD [41,42,43,44]. A recent meta-analysis confirmed the LUS score accuracy in early prediction of BPD in preterm neonates born below 32 weeks gestation [45].

### 3.6. Indication to Administer Surfactant 

Another intriguing application of LUS, is the aid to clinical decision-making in relation to the timing of surfactant administration. Surfactant administration based on the LUS score rather than a preset fraction of inspired oxygen (FiO_2)_, allowed for earlier surfactant administration, with reduction in oxygen exposure [46]. The application of the ESTHER policy (echography-guided surfactant therapy) obtained a higher amount of newborns receiving surfactant within the first 3 h of life, reducing oxygen exposure, decreasing the duration of invasive ventilation and increasing the number of ventilator-free days [47]. The global need for surfactant did not significantly change.

The ESTHER approach represents an innovative strategy to base exogenous surfactant treatment on a more physiology-based rational than an arbitrary oxygen limit, which not only may be set too high or too low depending on the operator preference, but it could also be influenced by multiple non-pulmonary factors [39].

### 3.7. Detection of Lung Diseases

LUS provides important information orienting the differential diagnosis between many pathological processes involving the lung [18,48,49,50,51,52]. LUS findings often overlap and are not fully specific of a single pathological process. However, when combining the clinical picture and medical history, LUS interpretation offers a much deeper understanding of the cause of respiratory distress than a chest-X ray alone. Some of the typical images of neonatal lung diseases are reported in Appendix A.

## 4. Cardiopulmonary Ultrasound

CPUS, the combination of LUS and TnECHO, has been shown to improve diagnostic accuracy for the identification of acute respiratory failure etiology in adult patients, frequently guiding clinical diagnosis and/or treatment [53,54]. Evidence-based guidelines for the neonatal use of point-of-care ultrasound (POCUS) have been recently released [54]. However, the specific application of comprehensive CPUS to the cardiorespiratory management of sick neonates is still largely unexplored.

### 4.1. Lung Ultrasound Guided Recruitment

Open lung ventilation aims to optimize lung volume through lung recruitment strategies, reducing atelectotrauma and subsequent ventilator lung injury [55]. Recently, LUS has been evaluated as an additional tool to guide lung recruitment strategies during mechanical ventilation in the adult and pediatric population [56,57]. Direct monitoring of lung re-aeration during the recruitment maneuvers can be performed at the bedside [56,58] and the LUS re-aeration score has been shown to be consistent with the changes in the pressure-volume curves [56]. A recent randomized controlled trial demonstrated the beneficial impact of LUS in guiding elective lung recruitment maneuvers as opposed to the oxygenation-guided method, in a population of preterm infants before the first surfactant administration [59]. LUS-guided recruitment better mitigated lung inflammation and reduced duration of mechanical ventilation. LUS guided recruitment has been attempted on non-invasive ventilation as well [60]. LUS guided recruitment constitutes an act of integrated CPUS, as the assessment of the hemodynamic status and eventual stabilization are prerequisite for the success of LUS guided open lung ventilation.

#### 4.1.1. S-Pattern and D-Pattern

One of the major issues about lung recruitment is the possibility to early distinguish a recruitable from an unrecruitable lung. We recently described two patterns that may allow for the early detection of lung recruitability and unrecruitability (Figure 3). The sunray(S)-pattern [61] develops in the reopening parenchyma during lung recruitment. The S-pattern develops early during the maneuver (within four pressure steps) [61] and represents the sonographic counterpart of the anatomical distal lung unit aeration during lung recruitment. Therefore, the S-pattern can only be visualized in the recruitable lung, potentially identifying the neonates that can benefit from the recruitment maneuvers. We showed the association between the S-pattern and the success of the LUS-guided pulmonary recruitment in critically ill neonates suffering from severe respiratory insufficiency [62]. In the case of a non-recruitable lung, the increase in airway pressure during lung recruitment produced dead-space recruitment without alveolar recruitment, generating the dead pace (D)-pattern on LUS (Figure 3) [61,62]. This scenario was associated with a poor response to the lung recruitment maneuver. In the presence of the D-pattern or when the S-pattern does not appear within four pressure steps, it may be reasonable to interrupt the lung recruitment maneuver to avoid hemodynamic instability or pulmonary complications. The addition of the S-pattern and D-pattern in the lung recruitment protocols may improve the outcome [63]. The role of the S-pattern and the D-pattern needs to be validated in prospective randomized trials.

#### 4.1.2. Lung Ultrasound Targeted Recruitment Protocol

We recently developed a protocol for LUS guided recruitment (Appendix A), that is especially useful when an inhomogeneous lung is anticipated. The lung ultrasound targeted recruitment (LUSTR) protocol [64] is based on the development of the S-pattern within four pressure steps and on the LUS-guided patient positioning before recruitment to target the airway pressure towards the most derecruited areas in the attempt to avoid segmental lung oversistension (Figure 4). In a retrospective case-control study the LUSTR protocol showed a better success of the maneuver, as compared to the standard lung recruitment based on oxygenation changes [64]. There was also a trend towards a reduced incidence of air-leaks in the LUSTR group, despite reaching higher mean airway pressures during lung recruitment. The LUSTR protocol, based on early detection of lung recruitability and LUS-guided patient position, may improve the efficacy and safety of lung recruitment in critically ill neonates. The efficacy of the LUSTR approach has to be confirmed in prospective randomized trials.

### 4.2. Lung Ultrasound Guided Position and Postural Recruitment

Postural recruitment can be defined as an active reaeration of atelectatic lung tissue by intentional changes of a patient’s body position under constant ventilatory conditions. This practice has shown positive results in adult and pediatric patients undergoing anesthesia [65,66,67]. In the presence of a recent and small atelectasis without severe impairment of the ventilation perfusion (V/Q) ratio or moderate aeration loss (B2 pattern), postural recruitment may achieve good results [68,69,70], potentially better than an arbitrary choice of position [71].

### 4.3. Phenotyping the Acute Neonatal Respiratory Diseases near Term

The transient respiratory distress after birth near term gestation is usually called transient tachypnea of the newborn (TTN). TTN, also known as wet lung, is universally considered a consequence of delayed resorption of fetal lung fluid [72]. The classic diagnosis of TTN is a combination of non-specific respiratory signs and radiographic findings. Moreover, a chest-X-ray often misses the significant derecruitment of the dependent regions and does not allow for the assessment of the pulmonary vascular resistances [72]. Different studies evaluated the role of LUS in better investigating the transient respiratory distress syndrome in the first few hours of life. LUS allowed for the identification of forms of acute respiratory distress into respiratory distress syndrome (RDS), meconium aspiration syndrome (MAS) and wet lung [72,73,74,75]. Depending on the inclusion criteria, the different studies reported different incidences of these respiratory diseases. We recently showed, that adding the TnECHO evaluation to LUS further improves diagnostic precision. We found that 50% of the patients with transient respiratory distress, clinically and radiologically diagnosed with TTN, showed signs of increased pulmonary vascular resistances (PVR) at TnECHO, either combined with some lung involvement (lung consolidation, wet lung, RDS) or alone. In particular, in 16% of the cases with signs of increased PVR, LUS was normal and the increased vascular resistance was the only explanation to the respiratory distress. Combining LUS and TnECHO we found six different phenotypes of transient acute respiratory distress in the newborn, that we defined as acute neonatal distress (AND) phenotypes [76]. 

CPUS evaluation in patients with acute respiratory distress may improve diagnostic precision. A better understanding of the transient respiratory symptoms after birth may assist communication between the NICU team and the family, eventually reducing parental stress. The classification of the neonatal transient respiratory distress according to the CPUS phenotypes may be more appropriate, than the symptomatic definition currently in use. The term “TTN” may in fact be confusing as it literally refers to an ambiguous group of symptoms that are common to different forms of cardiorespiratory disorders. Although the general knowledge is that TTN refers to the delayed clearance of lung fluid during labor and delivery, under the umbrella of TTN, there are different cardiorespiratory forms, that would largely go unrecognized without CPUS evaluation.

### 4.4. Phenotyping the Chronic Respiratory Diseases

BPD, defined as oxygen dependency at 36 weeks post-menstrual age (PMA), is now universally recognized to be a symptom resulting from a variable involvement of various pulmonary compartments, causing different BPD phenotypes [77,78,79]. The gold standard tools to diagnose more precisely BPD phenotypes (i.e., computed tomography, cardiac catheterization, bronchoscopy) are either invasive, require radiation exposure, or are not accessible to most centers [79]. An integrated POCUS assessment, including TnECHO and LUS may constitute a safe, non-invasive, radiation-free, and accessible approach to classify BPD phenotypes and longitudinally assess preterm infants throughout their NICU stay.

BPD-PH is the most recognized BPD phenotype in clinical practice, as it is routinely identified with TnEcho at the bedside [80]. The use of LUS in the case of BPD-PH may allow detection of eventual lung atelectasis and guide the patient’s positioning, aiming to recruit the collapsed respiratory units and optimize pulmonary hemodynamics [81]. As previously mentioned, LUS though B-line distribution and pleural line interrogation, may nicely assess the lung interstitial space, distinguishing forms of pulmonary oedema from inflammatory interstitial lung diseases, and potentially guiding treatment (Appendix A). In fact, although the use of diuretics [82] and late steroids [83] to prevent BPD have proven to be ineffective, the possibility to select those infants that show sonographic features of pulmonary edema versus signs of interstitial disease may allow to target patients who could benefit from diuretic or steroid treatment at an early stage of the disease [84]. Moreover, LUS may guide the best timing for treatment and longitudinally monitor response to drug therapy.

### 4.5. Cardiopulmonary Ultrasound Targeted Treatment

#### 4.5.1. Cardiopulmonary Ultrasound Approach to Patent Ductus Arteriosus

As previously mentioned TnECHO alone does not seem effective in guiding PDA treatment [13]. LUS can evaluate lung content in neonates with PDA and predict hemodynamic changes in persistent PDA [85]. Although still not investigated, LUS may integrate the TnECHO evaluation in order to select patients that may benefit from early treatment, such as those infants suffering from significant pulmonary overflow leading to pulmonary congestion and atelectasis of the dependent regions. Future studies using CPUS to guide PDA treatment instead of TnECHO alone may pave the way to a novel and more effective approach to the timing of PDA treatment. 

Moreover, considering the importance of the optimization of patient positioning in adults suffering from pulmonary edema and respiratory failure [81], LUS may guide the patient position and mitigate lung de-recruitment in the dependent regions until ductal closure, potentially reducing post-ligation cardiac syndrome.

#### 4.5.2. Cardiopulmonary Ultrasound Approach to Pulmonary Hypertension

When dealing with infants suffering from severe PH, a comprehensive CPUS assessment may not only target specific cardiac treatment, as previously shown (Figure 1 and Appendix A, Appendix A), but it may also reveal the presence and the location of lung atelectasis. It is general knowledge that the higher the intra-thoracic pressure during mechanical ventilation, the higher the right ventricle afterload. Therefore, the high intro-thoracic pressure may lower the right ventricle ejection fraction and worsen the PH picture. However, adequate oxygenation is of utmost importance in critically ill patients with PH. Alveolar and arterial hypoxia, acidosis, and hypercapnia worsen pulmonary vasoconstriction [86], which in turn may exacerbate the right ventricle dysfunction. Hypoxemia and hypercapnia are causes of regional perfusion redistribution within the lung and can trigger local vasoconstriction of the lung. LUS guided postural recruitment, trying to avoid active recruitment with increased airway pressure, may improve the V/Q ratio, potentially reducing the need for pulmonary vasodilators and/or improving the efficacy of iNO, ultimately optimizing pulmonary hemodynamics [86].

#### 4.5.3. Cardiopulmonary Ultrasound Approach to Neonatal Shock

In adult and pediatric intensive care units, integrated CPUS has shown a significantly better performance than LUS alone in the diagnosis of acute respiratory failure, being particularly useful to distinguish cases of hemodynamic pulmonary edema and pneumonia, often changing the clinical approach and management [87]. The CPUS approach leads to a shorter duration of vasopressor support [88]. Recently, it has been shown that in neonates, the expertise in CPUS interpretation to guide neonatal shock can be achieved in a relatively short time [89]. However, it still has to be determined if the CPUS approach to neonatal shock can significantly improve the outcome.

#### 4.5.4. Cardiopulmonary Ultrasound Approach to Cardiac Arrest and Sudden Deterioration

CPUS aid in deteriorating patients is invaluable and it has been standardized [89]. In addition to the previously described functions, CPUS can easily document and help treat life-threatening events, such as pericardial effusion, pneumothorax and pleural effusion (Figure 5, Appendix A). Figure 6 offers a tentative CPUS approach to the deteriorating infant.

#### 4.5.5. Cardiopulmonary Ultrasound Approach to Neonatal Acute Respiratory Distress Syndrome

Acute lung injury/acute respiratory distress syndrome (ALI/ARDS) is a life-threatening condition known for a long time in the adult ICU. ALI/ARDS recognize an inflammatory origin, leading to increased pulmonary vascular permeability and loss of aerated pulmonary parenchyma [90]. The lung inflammation that causes ALI/ARDS can be triggered by several stimuli, the most common being sepsis, pneumonia, chest injury, toxic chemicals, drowning, acute pancreatitis and adverse reaction to a blood transfusion. In adult patients suffering from ARDS, CPUS has been shown to improve the LUS diagnostic accuracy [91]. In COVID-related ARDS, CPUS has been used as a prognostic and monitoring tool [92]. Neonatal ARDS (NARDS) has been described for the first time in 2017 [93]. NARDS is classified as direct or indirect, infectious or noninfectious, and perinatal (≤72 h after birth) or late in onset [94]. Although not yet studied, the use of CPUS may improve diagnostic efficiency in NARDS, guide mechanical ventilation and target therapeutic choices.

## 5. Conclusions

CPUS may offer a valuable aid to the neonatologist, being potentially superior to LUS or TnEcho alone. Large prospective randomized trials are needed to demonstrate that this practice can affect important outcomes, such as mortality, morbidity and duration of NICU stay.

## Figures and Tables

**Figure 1 children-10-00462-f001:**
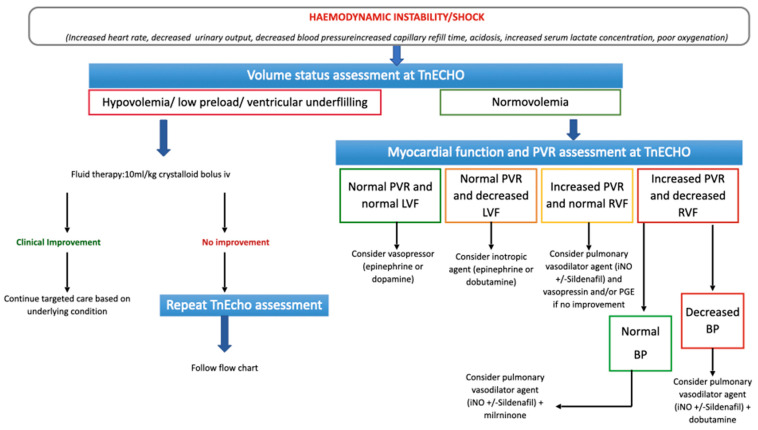
Targeted Neonatal Echocardiography (TnECHO) guidance for neonatal shock. Abbreviations: intravenous (iv); pulmonary vascular resistances (PVR); left ventricular function (LVF), right ventricular function (RVF); inhaled nitric oxide (iNO); prostaglandins (PGE), blood pressure (BP).

**Figure 2 children-10-00462-f002:**
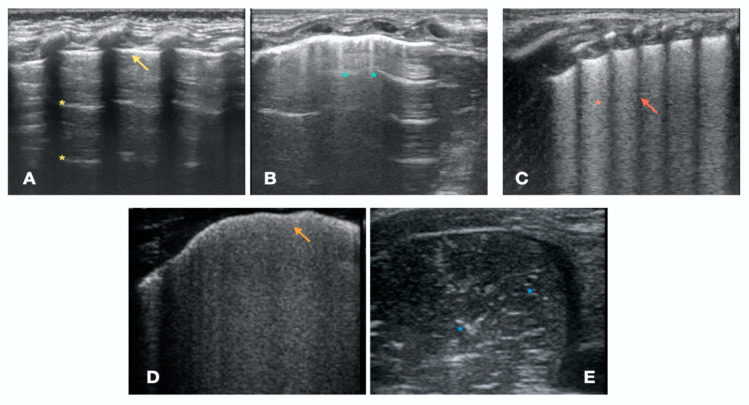
LUS patterns and artifacts. (**A**) A-pattern: normally aerated lung, with A-lines (yellow asterisks) and regular thin pleural line (yellow arrow). (**B**) B1-pattern, moderate loss of lung aeration: three or more B-lines (light-blue asterisks) occupying less than 50% of the scanned area. (**C**) B2-pattern: coalescing B-lines (red asterisk), occupying more than 50% of the scanned area, the acoustic shadow of the ribs is preserved (red arrow). (**D**) “white lung” with coalescing B-lines, erasing the acoustic shadow of the ribs and thickened pleural line (orange arrow). (**E**) C-pattern, complete loss of aeration (aeration score = three) with air bronchogram (blue asterisks).

**Figure 3 children-10-00462-f003:**
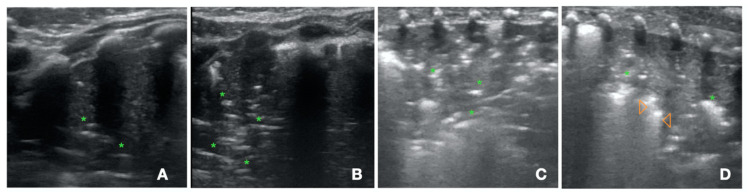
Dead-space (D)-pattern versus sunray (S)-pattern. (**A**) C-pattern with air bronchogram (green asterisks). (**B**) In this patient, the increase in continuous distending pressure (CDP) resulted in a more pronounced air bronchogram (green asterisks), suggesting an increase in dead space recruitment without any sign of alveolar recruitment (D-pattern). The D-pattern suggests a non-recruitable lung. (**C**) C-pattern with air bronchogram (green asterisks) (**D**) S-lines (orange arrowheads) develop from the reopening bronchi at the CDP increase, creating the S-pattern, that suggests lung recruitability.

**Figure 4 children-10-00462-f004:**
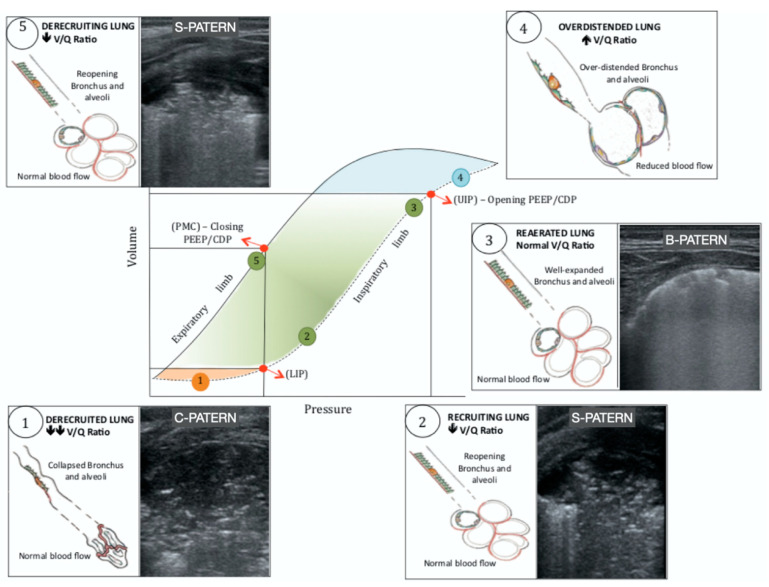
Lung hysteresis and lung ultrasound targeted recruitment (LUSTR). The lung hysteresis represents the difference between the pressure–volume (P–V) curve during inflation (dashed line) and deflation (continuous line). The lower inflection point (LIP) on the inspiratory limb represents the pressure below which lung de-recruitment (C-pattern on LUS) occurs (point 1). Below the LIP, the ventilation/perfusion (V/Q) ratio is decreased because of a reduction in ventilation with normal perfusion. In the recruitable lung, when lung collapse happens for any reason and a lung recruitment maneuver is attempted, a reopening sunray pattern (S-pattern) on LUS starts to appear right above the LIP (point 2). Then, pressure is increased until the S-pattern disappears and the B-pattern develops (point 3). This pressure corresponds to the opening positive end-expiratory pressure (PEEP)/continues distending pressure (CDP) at the upper inflection point (UIP) on the inspiratory limb of the P-V curve The UIP represents the point after which a further increase in airway pressure leads to over-distention without any improvement on lung recruitment. Above the UIP (point 4) the V/Q ratio is increased because of a reduction in blood perfusion due to the compression of lung capillaries by the over-distended alveoli. LUS does not distinguish hyper-expanded from normo-expanded lung. Once lung re-aeration is obtained, PEEP/CDP is reduced until an S-pattern, due to the initial lung de-recruitment, is detected (point 5). This is called closing PEEP/CDP and corresponds to the point of maximum curvature (PMC) on the expiratory limb of the P-V curve. In the ventilated lung, alveoli remain well expanded as long as PEEP/CDP is kept above the PMC and below the UIP. In this range of pressures, the V/Q ratio is optimal.

**Figure 5 children-10-00462-f005:**
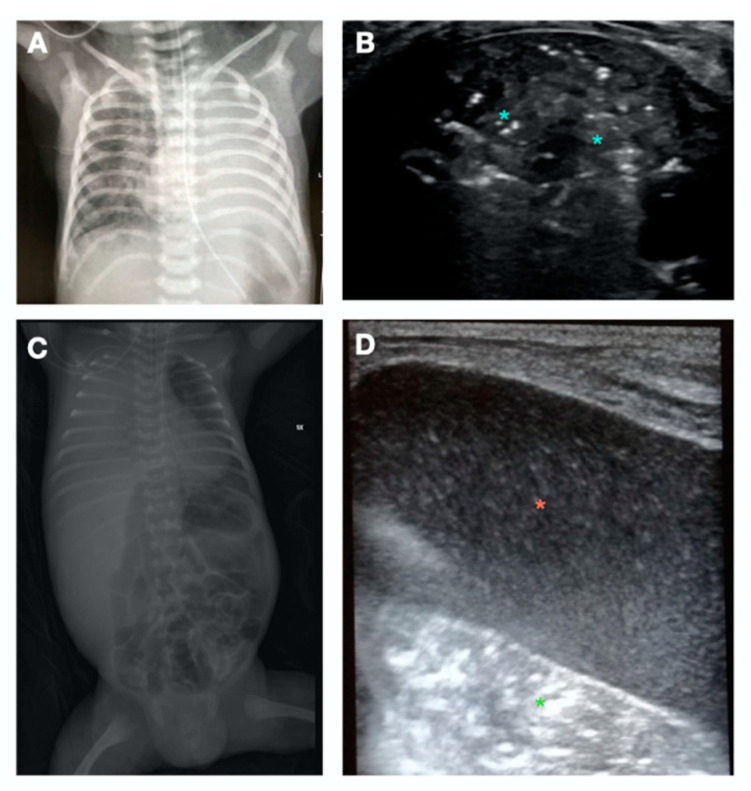
Detection of complications at CPUS. (**A**,**B**) Patient 1. (**A**) Chest X-ray from a patient with high oxygen requirement on mechanical ventilation showing left lung whitening. (**B**) Lung ultrasound (LUS) shows massive left lung consolidation with air bronchogram (light-blue asterisks). (**C**,**D**) Patient 2. (**C**) Chest X-ray from another patient with sudden deterioration showing right lung whitening. (**D**) Lung ultrasound (LUS) shows pleural effusion (red asterisk) with collapsed underlying lung (green asterisk).

**Figure 6 children-10-00462-f006:**
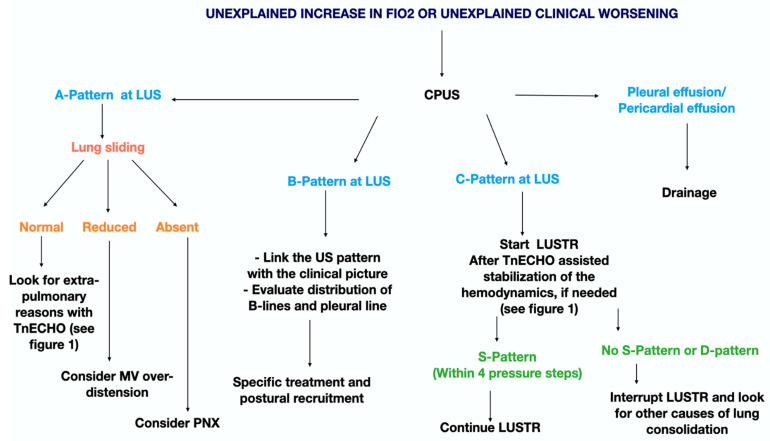
Cardiopulmonary ultrasound (CPUS) guided approach to the deteriorating sick infant. Abbreviations: fraction of inspired oxygen (FiO_2_), lung ultrasound (LUS), targeted neonatal echocardiography (TnECHO), lung ultrasound targeted recruitment (LUSTR), mechanical ventilation (MV), pneumothorax (PNX).

## Data Availability

Not applicable.

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
