# Peer review of "Use of Cardio-Pulmonary Ultrasound in the Neonatal Intensive Care Unit"

_children, 2023, doi:10.3390/children10030462_

Round 1

Reviewer 1 Report

General comment

This paper reviews the possible use of cardiopulmonary ultrasound in the neonatal intensive care unit. I think, this article will be useful for specialist doctors.

Specific points

1- I think, the authors should not use abbreviations at the headings and subheadings (e.g.)

2. TnECHO

2.2. PDA evaluation

2.3. Early and late PH

3. LUS

3.1. LUS technique and training

4. CPUS

2- I think, the authors should write “dysfunction” instead of “dysfuction

-Pulmonary overflow is due to increased pulmonary venous return which in turn causes left atrial volume overload and subsequent left ventricular dysfuction [7].

3- I think, the authors should write “BeNeDuctus trial” instead of “BeNeDucts trial

-The recent BeNeDucts trial showed that expectant management for PDA in extremely premature infants is non-inferior to early ibuprofen therapy administered according to pre-specified echocardiographic criteria of hemodynamic significance [13].

4- I think, the authors should not use abbreviations at the beginning of a line and of a sentence (e.g.)

-PH of the newborn is a condition characterized by increased pulmonary vascular resistances [14] that could affect full term newborns and prematures in different stages of their life with different pathogenesis.

5- I think, the authors should write “abnormal” instead of “abnoarmal

When considering abnoarmal vasoregulation the choice should be a vasopressor, such as dopamine, noradrenaline, vasopressin, terlipressin.

6- I think, the authors should write “to wean off” instead of “to wen off

LUS Scores can foresee readiness to wen off of NCPAP premature infants with evolving BPD [38].

7- The scientific names should be stated in the extended version the first time and then you can use the abbreviation (e.g). TTN, BPD, PMA

-As previously mentioned, TTN, also known as wet lung, is universally considered a  consequence of delayed resorption of fetal lung fluid[72] .

-BPD, defined as oxygen dependency at 36 weeks PMA, is now universally recognized to be a symptom resulting from a variable involvement of the pulmonary compartments, causing different BPD phenotypes [76-78].

8- I think, the authors should check the following sentence.

CPUS may offer a valuable aid to the neonatologist, being potentially superior to LUS or TnEcho e.

Author Response

General comment

This paper reviews the possible use of cardiopulmonary ultrasound in the neonatal intensive care unit. I think, this article will be useful for specialist doctors.

Specific points

1- I think, the authors should not use abbreviations at the headings and subheadings (e.g.)

2. TnECHO

2.2. PDA evaluation

2.3. Early and late PH

3. LUS

3.1. LUS technique and training

4. CPUS

A: we removed the abbreviations from the headings and subheadings

2- I think, the authors should write “dysfunction” instead of “dysfuction

-Pulmonary overflow is due to increased pulmonary venous return which in turn causes left atrial volume overload and subsequent left ventricular dysfuction [7].

A: We corrected the typo

3- I think, the authors should write “BeNeDuctus trial” instead of “BeNeDucts trial

-The recent BeNeDucts trial showed that expectant management for PDA in extremely premature infants is non-inferior to early ibuprofen therapy administered according to pre-specified echocardiographic criteria of hemodynamic significance [13].

A: we corrected the typo

4- I think, the authors should not use abbreviations at the beginning of a line and of a sentence (e.g.)

-PH of the newborn is a condition characterized by increased pulmonary vascular resistances [14] that could affect full term newborns and prematures in different stages of their life with different pathogenesis.

5- I think, the authors should write “abnormal” instead of “abnoarmal

When considering abnoarmal vasoregulation the choice should be a vasopressor, such as dopamine, noradrenaline, vasopressin, terlipressin.

A: we corrected the typo

6- I think, the authors should write “to wean off” instead of “to wen off

LUS Scores can foresee readiness to wen off of NCPAP premature infants with evolving BPD [38].

A: we corrected the typo

7- The scientific names should be stated in the extended version the first time and then you can use the abbreviation (e.g). TTN, BPD, PMA

-As previously mentioned, TTN, also known as wet lung, is universally considered a  consequence of delayed resorption of fetal lung fluid[72] .

-BPD, defined as oxygen dependency at 36 weeks PMA, is now universally recognized to be a symptom resulting from a variable involvement of the pulmonary compartments, causing different BPD phenotypes [76-78].

A: we stated the scientific names before the abbreviations

8- I think, the authors should check the following sentence.

CPUS may offer a valuable aid to the neonatologist, being potentially superior to LUS or TnEcho e.

A: we corrected the sentence

Reviewer 2 Report

line 28 write out Targeted neonatal ECHO

line 50 remove "to"

I like the description of PH, early and late but I'm unable to see table 2S and figure 2S to understand how reproducible line 84 is

uncertain what "good" pressure means line 90

line 97 "hypovolemia"

line 114 write out Lung Ultrasound

line 194 "wean"

line 205 remove second "to"

line 226 write out CardioPulmonary Ultrasound

line 315 "hysteresis"

consider placing link to technique videos early in paper

line 399 "specific"

I think pictures of A, B1, B2, C, S, and D patterns would be helpful interspersed within the text which describes them

Author Response

- line 28 write out Targeted neonatal ECHO

A: we wrote out Targeted neonatal ECHO

- line 50 remove “to"

A: we removed “to”

- I like the description of PH, early and late but I'm unable to see table 2S and figure 2S to understand how reproducible line 84 is

A: I put the supplementary material after the manuscript, so that it can be seen. I hope now it can be better evaluated 

- uncertain what "good" pressure means line 90

A: weremoved “good”

  • line 97 “hypovolemia"

A: we corrected the word

- line 114 write out Lung Ultrasound

A: we wrote out Lung Ultrasound

- line 194 “wean"

A: we corrected the typo

- line 205 remove second “to"

A: we removed the second “to”

  • line 226 write out CardioPulmonary Ultrasound

A: we wrote out CardioPulmonary Ultrasound

  • line 315 “hysteresis"

A: we corrected the typo

- consider placing link to technique videos early in paper

A: we put the videos earlier in the paper

- line 399 “specific"

A: we corrected the typo

- I think pictures of A, B1, B2, C, S, and D patterns would be helpful interspersed within the text which describes them

A: I put the supplementary material after the manuscript, so that it can be seen. I hope now it can be better evaluated